# Peptides with Dual Antimicrobial–Anticancer Activity: Strategies to Overcome Peptide Limitations and Rational Design of Anticancer Peptides

**DOI:** 10.3390/molecules25184245

**Published:** 2020-09-16

**Authors:** Yamil Liscano, Jose Oñate-Garzón, Jean Paul Delgado

**Affiliations:** 1Research Group of Chemical and Biotechnology, Faculty of Basic Sciences, Universidad Santiago de Cali, 760035 Cali, Colombia; jose.onate00@usc.edu.co; 2Research Group of Genetics, Regeneration and Cancer, Institute of Biology, Universidad de Antioquia, 050010 Medellin, Colombia; jean.delgado@udea.edu.co

**Keywords:** anticancer, peptide, *in silico*

## Abstract

Peptides are naturally produced by all organisms and exhibit a wide range of physiological, immunomodulatory, and wound healing functions. Furthermore, they can provide with protection against microorganisms and tumor cells. Their multifaceted performance, high selectivity, and reduced toxicity have positioned them as effective therapeutic agents, representing a positive economic impact for pharmaceutical companies. Currently, efforts have been made to invest in the development of new peptides with antimicrobial and anticancer properties, but the poor stability of these molecules in physiological environments has triggered a bottleneck. Therefore, some tools, such as nanotechnology and *in silico* approaches can be applied as alternatives to try to overcome these obstacles. *In silico* studies provide a priori knowledge that can lead to the development of new anticancer peptides with enhanced biological activity and improved stability. This review focuses on the current status of research in peptides with dual antimicrobial–anticancer activity, including advances in computational biology using *in silico* analyses as a powerful tool for the study and rational design of these types of peptides.

## 1. Introduction

Cancer is the second leading cause of death worldwide after heart disease, and the number of deaths is predicted to exceed 13 million by 2030 [1]. The traditional methods of treating cancer are surgery, radiotherapy, and chemotherapy; however, these methods are expensive, can often exhibit damaging effects on normal cells, and make cancer cells resistant to therapies and drugs [2].

Hence, the necessity to generate new drugs to combat problems of not only cancer drug resistance but also bacterial drug resistance, along with poor selectivity and adverse effects on normal cells has allowed peptides to make their way into the pharmaceutical industry [3,4]. Peptides emerged as one of the alternatives for therapeutic intervention because they mimic the natural metabolic effects within the body with low toxicity and better selectivity [5]. For example, peptides that can mimic hormonal activities are involved in various processes of antimicrobial defense, wound healing, and activation of immune response. To give a clearer example, we could talk about insulin, whose isolation, purification, and subsequent synthetic production have helped generations of people suffering from diabetes; adrenocorticotropic hormone (ACTH); and calcitonin [6].

Other peptides are oxytocin and synthetic vasopressin that paved the way for the synthetic biology market, which is always looking to decrease the use and damage of animals for the extraction and purification of molecules with therapeutic potential [7]. Many of these drugs were located within the “era of chemistry,” and are based on a rational design of compounds in terms of interaction with the target and the design of the ligand and the receptor [7].

In the 21st century, pharmaceutical companies experienced dramatic changes in drug safety regulations, lengthy compound development processes, and financial efforts that influenced investment in research and development [5]. Today, we could say that we are in the era of *in silico* design because of the tools of bioinformatics and molecular dynamics (simulation) that allow us to obtain information about the genome, transcriptome, and proteomics of organisms, thereby finding patterns and analyzing, modeling, and simulating molecules in systems similar to those presented by nature [7].

However, not everything is perfect with peptides as there are limitations to their use. These limitations include a short plasma half-life, which is due to the presence of peptidases that inactivate the peptides, and the possible immunogenicity that they can arouse in the host [8]. Another obstacle is oral bioavailability because when peptides pass through the digestive tract, specifically in the stomach, digestive enzymes disintegrate the peptide structure, affecting its biological activity [9].

Despite the disadvantages of peptides, to date more than 150 peptides in clinical trials have been approved in the United States, Europe, and Japan [6]. The current market for peptides is approximately USD 19 billion and by the end of 2020, it is expected to reach USD 23 billion [10]. To date, 63 peptides have been approved by the US Food and Drug Administration [10]. Peptides marketed include bacitracin, colistin, daptomycin, enfuvirtide, vancomycin, telavancin, teicoplanin, and dalbavancin [11].

The peptide therapeutics market is increasing year by year, starting with 10 peptides approved in 1980 to about 30 in 2010, with an average of 5 years in clinical studies before their release to the market [6]. This market is estimated at over USD 40 billion per year, equivalent to 10% of the pharmaceutical market [7]. There was a drop in the peptide market between 2010 and 2015 because pharmaceutical companies lost interest in peptides due to their low stability and oral bioavailability, replacing it with interest in small molecules that imitate peptides [6]. However, in recent years, the use of peptides has been on the rise because of advances in formulation, chemical modifications, and peptide delivery technologies [4], such as nanoemulsions, biopolymers (polyethylene glycol), and liposomes [5,6].

The current biopharmaceutical approaches are in search of new peptides from different species as there is an impressive variety, these peptides emerging from evolution mainly through the selective pressure, showing distinctive characteristics, such as selectivity and affinity against bacteria or cancer membranes [5]. One of the future challenges with peptides is to decrease protease degradation in our body through the use of liposomes or nanoparticles, improving bioavailability and peptide lifetime. Other challenges are improving in the solubility of most of the hydrophobic peptides, fast elimination, poor permeability of membranes, and cost of manufacture [11,12].

Nowadays, the interest in peptides is growing; the evidence of the growth is the rise in the number of publications on the topic of peptides. The most frequent publications on the Science Direct website are related to the topics of plants, mammals, and fish. The least frequent ones are about amphibians and frogs. However, authors, such as Uhlig et al. (2014) [5] point out that these publications are the most frequent in research, specifically on the topic of frog skin.

There are many areas of interest for various researches on peptides worldwide. However, the two aspects that are showing strength are the development of anticancer and antimicrobial peptides because of the resistance and low selectivity of conventional drugs [13,14]. There are two types of conventional anticancer drug resistance, intrinsic and acquired. Intrinsic resistance is related to genetic variations in the somatic cells of patients with tumors. Acquired resistance is due to the expression of energy-dependent transporters that eject anticancer drugs from the cells before they come into contact with the target [15].

In recent years, research has been conducted on a particular type of peptide that fights bacteria and cancer simultaneously. There are two types of anticancer peptides. The first one is active against mammalian, bacterial, and cancer cells. The second one corresponds to those active against bacterial and cancer cells, such as cecropins and magainins [16], and these are referred to as peptides with dual antibacterial–anticancer activity [3,17,18,19]. So, far there have been few studies on the physicochemical properties, structure, and characteristics of these peptides, allowing both bioprospecting and *in silico* design [3,17,19,20,21]. The study by Felicio et al. (2017) aimed to reveal certain characteristics of these peptides, such as alpha-helix and β-sheets structures, positive net charge, high hydrophobicity, and lengths of up to 30 residues [3].

In this review, we focused on the efforts to overcome the limitations of anticancer peptides. Then, we studied peptides with dual antimicrobial–anticancer activity and the rational or *in silico* design of this type of peptide.

## 2. Efforts to Overcome Peptide Limitations

The challenge of overcoming the limitations of peptides has driven the scientific community to search for solutions, such as peptide engineering via amino acid substitution, peptide conjugation, new formulations, and alternative delivery strategies [10]. Peptide engineering is based on the replacement of amino acids with d-amino acids to improve stability and half-life, to which N-acetylation and C-amidation are applied [22].

Peptide conjugation consists of peptide sticking, that is, the addition of polymers, such as poly(lactic-co-glycolic acid), polyvinylpyrrolidone, and polyethylene glycol, to peptides, which contributes to an improvement in bioavailability and an increase in half-life. It can also help prevent immune response and protect against degradation [10,23].

Another conjugation of peptides is with lipids, which is also known as lipidation. It produces amphiphilic peptides with increased bioavailability, improving the half-life, receptor selectivity, potency, and membrane penetration [24]. Daptomycin is one of these lipopeptides, which is produced from natural lipidation and is widely used to treat diseases caused by Gram-positive bacteria. Another example of natural lipidation is surfactin, which exhibits antimicrobial, antiviral, and antitumor activities [22,25]. Some of the peptides on the market that have gone through synthetic lipidation are liraglutide (Victoza^®^) and insulin detemir (Levemir^®^) [24].

Cycling, N-methylation, and lactamic bridges have improved the permeability, stability, potency, and solubility of peptides [22]. An alternative use of peptides to improve its potency and selectivity is the synergy of peptides with each other and with other conventional drugs. One example is the study by Kampshof et al. (2019) that shows how peptides, such as melimine and protamine, when combined with cefepime and ciprofloxacin, reduce resistance to fluoroquinolones in *Pseudomonas aeruginosa* (Figure 1) [26].

New formulation strategies to protect peptides from degradation include the addition of protease inhibitors. The incorporation of salt, sugar, and heparin in the formulations improves bioavailability, solubility, and stability in vivo [22]. Another type of formulation is hydrogels. They are three-dimensional fiber networks that can retain large amounts of water up to 1000 times their dry weight [28]. They can be made of homopolymers or molecules that can self-assemble into more complex structures [28]. These systems attract attention because of their varied applicability, such as drug delivery and tissue regeneration [29]. Peptide-based hydrogels, such as MAX8 (VKVKVKVK-VDPPT-KVEVKV-NH2) and RAD16 (AcN-RADARADARAD-CONH2), are remarkable because of their biocompatibility, biodegradability, and easy synthesis [10,29]. They also work as drug delivery matrices in a controlled way or as scaffolds to insert stem cells to promote tissue regeneration. Another advantage of these hydrogels is the ease with which they can be administered as they are injectable, thus avoiding surgical interventions [10,29,30].

Nanoparticles are another type of formulation used in inorganic and organic systems. In inorganic systems, nanoparticles are used with metallic ions, such as silver, gold, and zinc oxide. On the other hand, in organic systems, liposomes and polymer nanoparticles are found [31]. Similar to hydrogels, nanoparticles have high biocompatibility and biodegradability, and low cytotoxicity. Homopolymers and copolymers of polylactide or polyglycolic acid are frequently used. They are classified according to their architecture into nanospheres, nanocapsules, conjugated polymers, and polyelectrolyte complexes [31]. Nanoparticles can be used to transport proteins and peptides, whose conjugation generates a synergistic effect that improves the limitations of each one of the materials and achieves uses, such as inhibition of the interactions of pathogenic proteins and high sensitivity in molecular imaging [32,33,34].

## 3. Peptides with Dual Antimicrobial–Anticancer Activity

Chemotherapy remains the most widely used treatment for cancer. However, low selectivity is still a problem that could be improved by the use of peptides with anticancer activity. Moreover, these peptides are ideal drugs due to their decreased production cost (peptide synthesis reactors that can handle large amounts of reaction material for solid-phase peptide synthesis, substantially reducing costs [35]), easy chemical modification, and high capacity to penetrate tissues [36,37].

Antimicrobial peptides with anticancer activity are recognized as amphipathic cationic peptides that bind to and destroy cancer cells, either directly or indirectly [38]. The direct way is through the interaction with phosphatidylserine (POPS), an anionic phospholipid that is expressed more frequently in cancer cells. Once administered, electrostatic interaction occurs, opening pores that permeate the intracellular components, causing cell necrosis. The indirect mechanism is through the entry of the peptide into the cell without disturbing the membrane, inhibiting processes of protein synthesis, or damaging the mitochondrial membrane, which results in the activation of the apoptotic pathway mediated by caspases [38].

Many anticancer drugs destroy primary tumors along with their metastases without causing damage to normal tissues [39]. Activation of an antitumor immune response caused by these peptides has also been observed [40], and authors, such as Camilo et al. (2014), mention that it is very difficult to develop resistance against these peptides because anticancer peptides do not have a unique way of acting [41].

One of the future perspectives on anticancer peptides is the combination of these with chemotherapeutic drugs, thus improving the efficiency of destroying tumor cells with reduced toxicities and low risk of tumor recurrence [38]. The tested anticancer peptides include Magainin II, Buforin IIb, and BR2 as shown in Table 1.

According to the study by Felicio et al. (2017), the peptides found in trials until 2017 were ITK-1 in clinical phase III, Oncopore in phase II, and CLS-001 in phase II [3]. More current publications, such as Chiangjong et al. (2020), mention that MUC-1 is found in early phase I of the GAA/TT peptide. RNF43 721, NY ESO 1b, and HPV16 E7 are in Phase I. Synthetic human papillomavirus, WT1 peptide 126–134, and G250 peptide are in phase II. Leukemia peptide vaccine, PR1, is in phase III and degarelix in phase IV, which binds to gonadotropin releasing hormone (GnRH) receptors and blocks interaction with GnRH [42].

The details of tests done on peptides with dual antimicrobial–anticancer activity are given in Table 2. Some of these tests are antimicrobial activity, hemolytic activity, cytotoxicity tests on tumor cells, live imaging, Western blotting, DNA fragmentation tests, terminal deoxynucleotidyl transferase dUTP nick end labeling (TUNEL) assay, anti-angiogenesis assay, flow cytometry, release of lactate dehydrogenase (LDH), reactive oxygen intermediates (ROS) assay, analysis of morphological changes by hematoxylin-eosin (H/E) staining, and P-glycoprotein (Pgp) sensitivity assay [45,46,47,48,49,50,51,52,53].

## 4. Toward Rational Peptide Design

The discovery of new molecules in pharmaceutical development faces challenges as it is often a long and complex process with many faulty candidate molecules. The most frequent failures include lack of efficacy, undesired effects, and poor pharmacokinetics, which is due to the low bioavailability caused by the rapid degradation of peptides by proteases [54]. Moreover, many modifications and unnatural amino acids are also added, which decrease the stability of the peptides (stability is important for the peptide to retain its structure and carry out its functions) [4]. Another problem is the increased cost of making multiple modifications to a peptide to find the one with the best activity with the highest stability and low toxicity [54,55].

The identification of the target molecule has an estimated cost of USD 165 million, validation USD 205 million, optimization USD 120 million, and 75% of the costs are due to failures throughout the process, this process taking 10 years and only 1 in 12 molecules enter clinical trials [54,56].

The experimental identification and characterization of novel anticancer peptides is time-consuming and labor-intensive; therefore, there is a need for prior analysis to reduce the time, manpower, and cost of production, which plays an important role in preclinical evaluations of its toxic effects [56,57]. For example, in the pharmaceutical industry, when a compound is obtained and there is a need to improve certain characteristics, many chemical alterations are made that must be tested employing biological tests. This number of molecules can be reduced if a few are previously selected using *in silico* analysis. It means that these analyses serve to complement the biological results, and are currently used to identify a pharmacological target, lead, or active compound, and in preclinical tests [55]. *In silico* process will play a very important role in the discovery and development of new molecules that become a competitive advantage among pharmaceutical companies and will make the difference between those who apply them and those who do not [54].

This crucial role is currently being played by bioinformatics together with omics, which are the tools that allow the study of a large number of molecules involved in the functioning of an organism [54]. New sequencing technologies began to appear that managed to reduce the time in sequencing genomes, reducing costs with high quality [54]. Then, new omics, such as transcriptomics, proteomics, metabolomics, phenomics, and metagenomics arrived, filling biological databases, such as NCBI PubMed with information [58]. This database includes a wide variety of information, for example it is a database of genes, proteins, genomes, and even transcriptomes [58].

As sequencing costs have been reduced, biological information has increased and new disciplines, such as data mining have emerged to find patterns and create predictive models [59]. With these models, it is possible to understand phenomena, such as knowing the type of amino acid that is most likely to form an alpha-helix or β-sheets structure. This information is useful in building three-dimensional models of the sequences that make up a peptide [60].

In the past decades, artificial intelligence has been added to these predictive models along with an area of knowledge known as machine learning (ML), which consists of the study of computer algorithms that learn with experience and identify patterns or trends presented by data. The supervised algorithms are divided into two groups, regression and classification. The unsupervised algorithms are also ordered into two groups, clustering and dimensional reduction [58,59]. The monitored algorithms work with pre-tagged and known data to predict the output values of a function [58]. On the other hand, unsupervised algorithms are used for classification problems, them to make diagnoses, and widely in the banking field for identity fraud detection [61].

Regression is also used to predict life expectancy, population growth, and weather forecasts [58]. However, it is noteworthy that for better prediction, a large volume of data is needed, and these data should be of good quality; the focus of ML is to find the best model that fits the data. This model is contrary to the traditional model where the data fit the model. In the latter case, it is important to consider filtering outliers and incorrect data [61].

All the above-mentioned processes have allowed the opening of a new field in peptide research, the *in silico* or rational design of peptides. This is based on three main approaches: the physicochemical approach, which is based on the modification of a sequence according to its physicochemical parameters to improve its stability, selectivity, and potency by making additions, substitutions, or elimination of residues, as well as reducing the size of the sequence and adding functional groups as mentioned above through peptide engineering. The second approach in rational peptide design is the use of templates, which can be proteins, other peptides, or can also be obtained by genome, transcriptome, or proteome alignments using reference peptide sequences. The last approach that is gaining more strength in the *in silico* design of peptides is the generation automated computational methods to enable fast and accurate prediction, screening and design of novel anticancer peptides based on the application of predictive models, using ML algorithms and neural networks [57,62]. These approaches can be combined to generate a hybrid rational design [57,63].

## 5. Physicochemical Methods of Rational Design of Anticancer Peptides

It is well known that changes in the physicochemical parameters have been the most frequent method for the rational modification of anticancer peptides, either by increasing or decreasing mainly the net charge, percentage of hydrophobicity, amphipathicity, and helicity that allow the ability of these peptides to interact with cell membranes or to improve the stability and selectivity [15,56,64].

The net charge is important in the design of antimicrobial peptides because they allow the electrostatic interaction of peptides with the bacterial cell membrane. Therefore, cationic type of peptides are those that present positive charge and are able to interact with phospholipids, such as POPS and phosphatidylglycerol that present negative charge in the polar head [65]. POPS is also present in the mitochondria and cardiolipin, allowing peptides that achieve internalization and bind to the mitochondrial membrane to destabilize it by cytochrome-C release with consequent activation of apoptosis [66].

Gaspar et al. (2013) mention that there are other negative charges on cancer cell membranes, such as O-glycosylated mucins, heparin sulfate, and sialylated gangliosides [16]. It has been found that sialylation in the cancer cell membrane is a determining factor in the interaction with anticancer peptides, varying their content among cancer cells as shown by Miyazaki et al. (2012) with Magainine II, whose affinity for binding to gangliosides increased in the presence of sialic acid [67]. Risso et al. cleaved sialic acid from the surface of U937 lymphoma cells and found reduced activity of the BMAP-27 and BMAP-28 peptides [68].

Cationic peptides with charges between +2 and +9 usually have better interaction with the anionic heads of phospholipids [69,70]. Above +9, peptides tend to develop hemolytic activity which counteracts their clinical use [71]. Moreover, the electrostatic interactions are stronger and prevent the peptide from inserting into the membrane and deforming its helical structure, thus, losing antimicrobial activity [64,72]. Although anionic peptides with anticancer activity are found, cationic peptides are widely used considering their cytotoxicity through the lysis of the cancer cell membrane. Another mechanism used by cationic peptides different from membrane lysis is the ability to incorporate themselves inside the cell, acting as cell-penetrating peptides called tumor-homing peptides [10,66]. This particular class of peptides induces activation of endocytic pathways related to macropinocytosis through binding to tumor cell receptors, such as neuropilin, which are responsible for activating these pathways [66]. The motif in some of these peptides is CendR (R/KXXR/K) (arginine/lysine, two hydrophobic amino acids, and arginine/lysine), which interacts with neuropilin-1 and allows the internalization of the anticancer peptide [10].

Another type of peptide designed to destroy cancer cells is the hunter–killer peptide (HKP) that are short chimeric molecules (∼20 amino acids), which act as a ligand for receptors on the target cells and induces cell death via disruption of mitochondrial membranes with cytochrome-C release, caspase activation, and apoptosis [73,74]. These peptides consist of two fractions coupled by a linker often glycine–glycine; the first fraction (hunter: 5–10 amino acids) is designed to bind to the target cell receptor and the proapoptotic domain (killer: ∼14 amino acids) is responsible for inducing apoptosis [73,74]. The HKP-1 is targeted to the angiogenic vasculature of tumors and has strong anticancer activity in models of breast and prostate cancer, reducing tumor volume and metastasis and prolonging survival [73]. Another type of peptide conjugation was carried out in the work of Almaaytah et al. (2019), using an enzyme-based cleavage strategy with an anticancer peptide, cytropin A, which is a hybrid peptide resulting from the union of the cytropin 1.1 (it has low selectivity) and a matrix metalloproteinase (MMP) consensus sequence that would be cleaved to release the active cytropin once it meets highly metastatic MMP producing cancer cells, as highly invasive tumors secrete a large amount of the metalloproteinase matrix responsible for metastasis [75].

The amphipathicity of the antimicrobial peptides is differentiated in two, hydrophobic and hydrophilic regions, each facilitating the electrostatic and hydrophobic interactions between the peptide and the membrane with the anionic heads and the hydrophobic chain of the phospholipids [70,76]. This property is measured through the hydrophobic moment; the higher this value, the more the antimicrobial activity increases [77,78]. The parameter used for hydrophobic interactions of the membrane with the peptide is the hydrophobic percentage whose optimal range of potent antimicrobial activity is between 40% and 50% [69,72]. It can be used to increase peptide potency without sacrificing selectivity [79]. According to studies by Yang et al. (2013) using temporin-1CEa, increasing net charge while reserving the moderate hydrophobicity may be a strategy to improve the cytotoxicity against tumor cells and decrease the hemolytic activity [80].

Hydrophobicity is also important in the mechanism of action against cancer cells, allowing anticancer peptides to penetrate the membranes [81]. Huang et al. (2011) found that increasing hydrophobicity of the V13K peptide via substitution of A12L and A20L causes a greater effect against human cervix cells; however, selectivity decreases, hence V13K also attacks normal cells [82]. It has been observed that changes in hydrophobia using W-tagging provide a powerful and broad antimicrobial spectrum and increase the probability of being internalized and generate toxicity against melanoma cells [66].

Helicity is another characteristic of peptides with dual activity. It has been shown that an increase in helix propensity increases the potency [63,83]. Huang et al. (2012) modulated the helicity of 26-residue amphipathic α-helical peptide A12L/A20L (Ac-KWKSFLKTFKSLKKTVLHTLLKAISS-amide) by introducing d-amino acids to replace the original l-amino acids on the non-polar face or the polar face, finding a strong correlation between helicity and hemolytic activity of peptides. They also observed that a lower helicity decreases the cytotoxic activity on Hela cells [83]. Another structure present in peptides with dual antimicrobial–anticancer activity is β-sheets; however, they are only about 2% of this type of peptides [15]. Among the peptides with this structure is SVS-1 (KVKVKVKVDPLPTKVKVKVK-NH2), which folds only at the surface of cancer cells and acquires a β-sheets structure that disrupts the cell membrane via pore formation [16,84].

The secondary structure affects not only potency and selectivity of the peptides, but also their structural orientation as, depending on the angle of orientation, the peptide produces destabilization of the membrane phospholipids and affects permeability [44]. Peptide length is also a feature to consider in the design of anticancer peptides because short peptides have been found to have low immunity, low production costs, shorter synthesis, and shorter production time [85].

Other important properties to consider are stability and peptide aggregation. There have been many efforts to improve the stability of peptides concerning interactions with the environment and molecules, such as proteases, developing vehicles for their transport as liposomes and lipidic nanoparticles, such as solid lipidic nanoparticles and nanostructured lipidic transporters [86,87]. Its importance lies in the fact that the therapeutic effect of the molecule can be reduced by changing its structure [88]. However, it is not yet clear how structural stability influences the potency of antimicrobial cationic helical peptides [89].

The potency also relates to stability with flexibility, which is another structural determinant of antimicrobial activity. This flexibility is reflected as a hinge near the central position of a chain α-helicoidal, which allows the peptide to cross the lipid bilayer and play important roles in bacterial cell selectivity and antimicrobial and antitumor activities [90]. The B-factor or temperature factor for structure determination by X-ray crystallography, can be used to evaluate protein flexibility, thermal stability, and intrinsic disorder, with flexibility a dominant determinant of activity and it should be useful to look for a new structure–activity relationship for a cationic antibody peptide α [90].

Antimicrobial peptides have a propensity for intermolecular interactions (hydrogen bridges, electrostatic forces, hydrophobic interactions) leading to oligomerization and aggregation [91]. It has been proven, through molecular dynamic simulations and experimental assays, that the increase in intermolecular interactions through interpeptide aggregation increases the cost of energy for the peptide to embed itself in the bacterial cell membrane, this, in turn, decreases the antibacterial activity [91].

Once the changes at the level of physicochemical parameters to improve peptide activity and stability are made, the prediction of the three-dimensional structure is elaborated using modeling tools to finally carry out molecular docking and/or simulation processes through molecular dynamics to predict peptide behaviors in systems, such as membranes, and/or to help interpret in more detail the biological results [56]. These processes are known as *in silico* design because they are done through the computer by performing computer simulations [55].

## 6. Sequence Template Methods of Rational Design of Anticancer Peptides

The sequence template methods refer to taking a known peptide sequence as a basis to reduce its size or fragment the reference sequence and add modifications, such as amino acid substitution, fragmentation, cycling, hybridization, and so on, allowing improvement of the stability, potency, and selectivity of anticancer peptides as well as the alteration of the physicochemical properties [63,85]. An example of hybridization is the study by Hao et al. (2015) who hybridized the peptide HPRP-A1 (FKKLKLFSKLWNWK) with the peptide TAT (RKKRRQRRR) forming the peptide HPRP-A1-TAT(FKKLKLFSKLWNWKRKKRQRRR) with improved anticancer potency [85,92].

Fragmentation is a type of alteration of the original sequences to shorter sequences; these original sequences can be proteins with known anticancer activity, for example, the anticancer peptide EMTPVNPG obtained from the alpha fetus protein, or proteins without anticancer activity reported in the following case of the peptide HPRP-A1 derived from the N-terminus of ribosomal protein L1 of *Helicobacter pylori* [85]. Hu et al. (2016) showed us how a peptide with anticancer activity known as buforine can be obtained from histone H2A [85]. Anticancer peptides with improved selectivity can also be obtained from the fractionation of other peptides, such as buforine IIb, deriving from this the BR2 peptide with a length of 17 residues without toxicity to normal cells as presented in Lim et al. (2013) [93]. Furthermore, the 10 amino acids at the N-terminal end of cecropin can be used and repeated three times to create the CB1 peptide [85]. Grissenberger et al. (2020) obtained anticancer peptide fractions from lactoferrin residues 21 to31, including retro dipeptides DIM-LF11-322 and R-DIM-PLF11-215 characterized by a net charge of +9 and a potent activity and selectivity against cancer cells [94].

Karbalaeemohammad et al. (2011) obtained the dual antimicrobial–anticancer activity peptide, TempY (FLPLIGKLLSGLY-am), from the template aurein 1.2 whose sequence was designed by the Rosetta Design server (http://rosettadesign.med.unc.edu/), using an amino acid pattern H H P H G K H H S G HH (H denotes hydrophobic and aromatic amino acids, P denotes proline, and S and G denote serine and glycine amino acids), which the authors claim allows for antimicrobial activity and by adding more lysines to create anticancer activity as well, based on the fact that increasing the positive charge on the peptide sequence will enhance the interaction with the negative charge of the cancer cell membranes [95].

Template methods can also be used for the analysis of molecular docking with proteins related to cancer, such as heat shock protein 90 alpha (Hsp90a), which regulates oncoproteins. Therefore, the inhibition of this protein is a therapeutic target against breast cancer according to the work of Gupta et al. (2013); they hypothesized that Hsp90a interacts with Hsp organizing protein (HOP) helping the interaction with Hsp70 to work properly. They performed the docking (software Hex 6.1, Nancy, France; http://hex.loria.fr/) between Hsp90a and HOP and then the residues interacting with the active site were selected for the design of new peptides using PEP-FOLD, which is a program for de novo 3D modeling. Afterwards, with the structure of the peptides, the docking was done and finally, they obtained the best Hsp90a inhibitor with high binding energy, less amyloidogenic properties, and high solubility called PEP73 [96]. Moreover, using Hsp90 as a target, Plescia et al. (2005) designed shepherdin using structure-based mimicry to interrupt survivin–Hsp90 interaction, first evaluating its *in silico* activity through docking and molecular dynamics to move on to in vitro and in vivo analyses, finding that shepherdin is capable of inducing massive tumor cell death by apoptotic and non-apoptotic mechanisms without reducing the viability of normal cells [97].

Peptides can also be predicted from alignment to peptide databases, such as E-Kobon et al. (2016) who present a combination of proteomics and bioinformatics using the *Achatina fulica* mucus also known as the African snail, obtaining the F2 and F5 fractions after a process of purification and sequence determination. These fractions were aligned against the CancerPPD database to obtain putative anticancer sequences. Then, they were subjected to the prediction of anticancer activity with two programs, ACPP (http://acpp.bicpu.edu.in/predict.php) and AntiCP (http://crdd.osdd.net/raghava/anticp/), using the support vector machine (SVM) learning algorithm. ToxinPred (http://crdd.osdd.net/raghava/toxinpred/) was used to predict the toxicity and CellPPD (http://crdd.osdd.net/raghava/cellppd/) was used to predict the cell-penetrating activity [98].

In the study by Li et al. (2018), the Gonearrestide peptide derived from scorpion venom was identified using a high-throughput platform combining transcriptome and proteome sequencing employed successfully to enable large-scale, high-throughput identification of novel bioactive peptides in venoms. The transcriptome was assembled de novo with Trinity (https://github.com/trinityrnaseq/trinityrnaseq/releases). With the software PEAKS (version 8.0, Bioinformatics Solutions Inc., Waterloo, ON, Canada; https://www.bioinfor.com/), candidate peptides were obtained to eliminate the already reported sequences and keep the new ones. The candidates were aligned using blast and after a series of bioinformatic analyses together with in vitro functional biological examinations, it was discovered that Gonearrestide is a very potent anticancer peptide that acts on a wide spectrum of human cancer cells and causes little or no cytotoxic effect in epithelial cells and erythrocytes [99].

Finally, in template methods, we cannot forget amino acid modification (addition, removal, or change) that is important in terms of their interaction with the membrane, DNA, and proteins of the cancer cells [44]. Positively charged residues, such as arginine, lysine, and histidine tend to disrupt cell membrane integrity and induce cytotoxicity in cancer cells via membrane permeability [67,100,101]. Tumor tissue is characterized by a lower pH (6.2–6.9) than normal tissues (7.3–7.4), and increased activity of histidine-rich peptides has been found at reduced pH [67,102]. Negatively charged amino acids, such as aspartic acid and glutamic acid have antiproliferative activity on tumor cells [103]. Proline, tyrosine, and phenylalanine are characterized by their interaction with the phospholipids of the cancer cell membrane and increase the cytotoxicity of anticancer peptide [81,104,105]. It has also been observed that proline insertion tends to reduce helicity, similarly to D-amino acids, leading to reduced activity [63]. Proline-rich peptides, such as p1932 (NH_2_-GPPPQGGNKPQGPPPPGKPQ) achieved internalization into squamous cancer cells [106]. Reduced methionine will stop the proliferation of cancer cells [44], and tryptophan enters these cells via the endocytic pathway and binds to the major DNA groove [107,108]. Peptides with tryptophan increase cytotoxicity against non-small cell lung adenocarcinoma A549 cells but hemolytic activity also increases according to the Conlon study et al. (2013) [109].

## 7. Automated Computational Methods for Anticancer Peptide Prediction

Automated computational methods emerge as an alternative to the time-consuming and costly process of screening peptides with a probable anticancer activity that can take months to years with a high risk of failed molecules [110]. Automated computational methods are based on the use of ML approaches and deep learning (DL), which is a field of artificial intelligence that automates analytical model building for rapid and accurate outcome prediction, observing an increase in the number of ML-based anticancer prediction tools, such as AntiCP, where the scientific community has free access to create mutant sequences with probable anticancer activity [57,111].

ML together with DL are becoming new resources for predicting both the secondary structure and function of peptides. ML and DL are being used to solve problems not only in the area of structural bioinformatics but also in social media services, online customer support, product recommendations, self-driving cars, translators, fraud detection, image recognition, and other applications [112].

These artificial intelligence algorithms discover patterns in the data to improve decisions or actions in complex applications with the being case in the determination of the secondary structure of peptides; subcellular location; peptide–protein, peptide–membrane, and peptide–DNA interactions, and folding from the sequence, as well as to predict their probable function. This information is then validated at the experimental level [113,114]. In this way, we can see how a new field of research and development of predictive tools is opening up using automated computational methods, which will improve the process of designing, selecting and evaluating drugs, such as antimicrobial–anticancer peptides.

Clustering by *K*-means and SVM has been extended to classify peptides in relation to their structure and function. Most of these prediction methods use an SVM-based *in silico* method, considering the composition of the amino acids and dipeptide and the binary discrimination of being anticancer peptides or not [57,110,111,114]. For the construction of the prediction model, first the sequences of the anticancer peptides or peptide library must be collected to have a dataset (the raw material for the Prediction System), which can be found in databases such as APD3 (http://aps.unmc.edu/AP/database/antiC.php), which so far has 249 anticancer peptides; DRAMP 2.0 (http://dramp.cpu-bioinfor.org/quick_search.php?srh_txt=anticancer), which has 293 entries and prediction tools, such as physicochemical property prediction and secondary structure prediction; CancerPPD (http://crdd.osdd.net/raghava/cancerppd/info.php), which has 3491 entries and information related to various chemical modifications, such as non-natural, d-amino acids, and modified amino acids (ornithine). TumorHope has 744 entries of experimentally characterized tumor-homing peptides (https://webs.iiitd.edu.in/raghava/tumorhope/) [114,115,116,117].

The next step is to choose the features (attributes that describe each instance of the benchmark dataset) of the sequences we want to use to classify the anticancer peptides such as the amino acid composition (AAC) and dipeptide composition (DPC). With these data, the ML-based prediction model is fed or trained to identify patterns and make the classification of the new data incorporated in the model [62,111,112,118]. After training with the dataset, the validation must be done to evaluate the accuracy of the prediction and its error. However, given the stochastic nature of both the construction of these data sets and the variables and parameters of the classification models used, the same results will not necessarily always be obtained and, consequently, the same precision and error will be obtained in each run of the algorithms. Thus, it is suggested to use cross-validation, which is a technique used to evaluate the results of a statistical analysis. It is used where the main objective is prediction and need to estimate the accuracy of a model ensuring that they are independent of the partitioning between training and test dataset [119,120].

Cross-validation consists of taking the original dataset and creating from it two separate datasets, 70% of this data will be the training dataset, the remaining 30% will be used as a validation dataset. Then, the training data will be divided into *n* subsets and, at the time of training, each *n* subset will be taken as a test set of the model, while the rest of the data will be taken as a training set. This process will be repeated *n* times and, in every iteration, a different test dataset will be selected, while the remaining data will be used, as mentioned, as a training set. Once the iterations are completed, the accuracy and error are calculated for each of the models produced [119,120].

There are three main cross-validation methods, which are often used to examine an anticancer peptide predictor for its effectiveness: the independent dataset test, subsampling test, and jackknife test, which is considered the most objective and has been used to examine the performance of several predictors, giving a single result for a dataset [121,122].

Based on the above-mentioned process, multiple prediction tools have been created, such as cell-penetrating peptide prediction (http://crdd.osdd.net/raghava/cellppd/) [118,123,124], antimicrobial peptide prediction (http://www.camp.bicnirrh.res.in/prediction.php) [125,126], anticancer peptide prediction (https://webs.iiitd.edu.in/raghava/anticp2/) [111], DL for protein secondary structure prediction from the primary sequences (DeepPrime2Sec) (https://llp.berkeley.edu/DeepPrime2Sec/) [127], and predicting therapeutic peptides by DL [128].

Below, we will see how anticancer peptide predictors are improving their models, for example, one of the first strategies to predict this type of peptide was carried out by Tyagi et al. (2013), who developed the AntiCP predictor using SVM-based models with AAC and DPC as input features with a maximum accuracy of 91.44% [129]. Nevertheless, the study by Vijayakumar et al. (2015) showed that there was no significant difference in AAC between anticancer and non-anticancer peptides, incorporating centroidal and distributional measures of amino acids [130]. Hajisharifi et al. (2014) proposed a model based on Chou’s pseudo AAC and the local alignment kernel-based method [131].

Chen et al. (2016) developed the iACP predictor using g-gap dipeptide component optimization approach to establish a really useful sequence-based statistical predictor, which improves the accuracy of the model compared to its predecessors. This development was possible by following Chou’s five-step guidelines: (1) benchmark dataset, (2) sample representation, (3) operation engine, (4) cross-validation, and (5) web-server [132,133]. Afterward, Manavalan et al. (2017) appear with the MLACP predictor, adding features of AAC, DPC, atomic composition, and physical–chemical properties to the SVM and random forest prediction algorithms. Unlike previous models that used the jackknife test for cross-validation, they adopted the 10-fold cross-validation to reduce computational time [122]. Wei et al. (2018) revealed a novel feature representation learning scheme to integrate the class information of data into features and effectively explore a set of more informative features. They employed a two-step feature selection technique that results in a five-dimensional feature vector with greater discrimination power than its predecessors to identify anticancer peptides. They also argue that the feature descriptors used to build the predictive models are based on the use of sequential information, as AAC does, and are not very informative to discriminate true anticancer peptides from non-anticancer peptides [131].

By 2019, three authors Boopathi et al. (2019), Schaduangrat et al. (2019), and Yi et al. (2019) appear with their respective proposals for anticancer peptide predictors, mACPpred, ACPred, and ACP-DL. Boopathi et al. used a 10-fold validation method unlike Schaduangrat et al. who used jackknife cross-validation and Yi et al. who used five-fold validation [1,134,135]. The new approach proposed by Boopathi et al. was to use seven feature encodings, including AAC, DPC, composition–transition–distribution, quasi-sequence-order, amino acid index, binary profile (NC5), and conjoint triad. They excluded irrelevant features by applying a two-step feature selection protocol and identified their corresponding optimal feature-based models and used SVM to build the model mACPpred Hydrophobic residues in an alpha-helix structure, cysteine residues in a β -sheets structure, and the formation of an amphipathic alpha-helix structure played a crucial role in the anticancer activity, according to the study by Schaduangrat et al. [134]. Yi et al. used the deep long short-term memory model approach to predict anticancer peptides by representing high-efficiency features called ACP-DL, this model was compared with those of SVM, random forest, and Naive Bayes with good results [135]. Ge et al. (2020) proposes a new two-step learning model for the identification of anticancer peptides (EnACP). They used feature representation, composed by AAC, autocorrelation, pseudoAAC, and profile-based features generating 19 kinds of feature patterns, which were first classified using light gradient boosting machine. Then, the predicted results were the input into an SVM classifier to obtain the final prediction [136].

In summary, the increase in the number of anticancer peptides that are obtained at the experimental level and through rational design has led to an increasing refinement of ML–DL-based anticancer peptide prediction models; however, they need to be further developed to obtain high accuracy. Therefore, the continuous improvement of these models strengthens the *in silico*–experimental relationship for the search of new molecules in a faster, more efficient, and low-cost way.

## Figures and Tables

**Figure 1 molecules-25-04245-f001:**
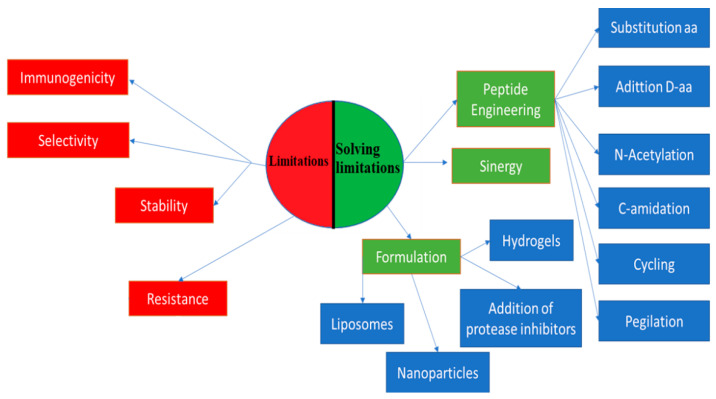
Peptide limitations and how to overcome them. [10,22,23,26,27]. On the left side there are four limitations to working with peptides which are in red. In green on the right side the three main strategies to mitigate the limitations are shown and also the more specific strategies broken down in each of them in blue.

**Table 1 molecules-25-04245-t001:** Anticancer peptides and examined cell lines [12,43,44].

Anticancer Peptides	Examined Cell Lines
Magainin II	Bladder cancer cells
Buforin IIb	Cervical carcinoma cells
BR2	Cervical carcinoma cells
PNC-2 and PNC-7	Pancreatic cancer cells
RGD-PEG-Suc-PD0325901	Melanoma A375 cells
p16	Pancreatic cancer cells
Defensin	Lung Carcinoma cells
LL-37	Ovarian Carcinoma, Breast Cancer cells
Cecropin A y B	Bladder cancer cells
Bac-7-ELP-p21	Ovarian Carcinoma cells
NRC-3 and NRC-7	Breast Cancer cells

**Table 2 molecules-25-04245-t002:** Test for peptides with dual antimicrobial-anticancer activity [45,46,47,48,49,50,51,52,53].

Test	Information
Antimicrobial activity	Used to find the Minimum Inhibitory Concentration and the Bactericidal Concentration that kills 99.9% of the bacterial population. At present, microdilution is frequently used in 96-well plates and the reading can be done visually or through the creation of a curve relating the percentage of inhibition by the peptide and the concentration.
Hemolytic activity	Used to find the hemolytic concentration 50, a useful parameter to determine the degree of cytotoxicity that the peptide can cause in eukaryotic cells.
Cytotoxicity test on tumor cells	This test is usually performed by screening with (3-(4,5-dimethylthiazolyl-2)-2,5-diphenyltetrazolium bromide) (MTT), this colorimetric test allows the evaluation of the cellular metabolic activity by reducing the MTT to its insoluble form formed by oxidoreductase enzymes, changing from yellow to purple with the appearance of the formazan in living cells.
Live imaging	For this test, the cell nucleus is marked with 4′,6-diamidine-2-phenylindol (DAPI) and the peptide with another marker such as Fluorescein isothiocyanate (FITC) and observed by fluorescence microscopy. By means of this test it is possible to have a vision of the mechanism of damage of the anticarcinogenic peptide.
Analysis of morphological changes by H/E staining	The cells are fixed with methanol for 1 min and stained with H/E to visualize the morphology of the cells.
Pgp sensitivity assay	Pgp is a drug transporter that plays important roles in multidrug resistance and drug pharmacokinetics. The inhibition of Pgp has become a notable strategy for combating multidrug-resistant cancers.
Western blotting	It is used to determine if there is caspase activation or not and also to determine whether the peptide damage was caused by necrosis or apoptosis. To determine apoptosis, antibody against caspase 3 is incubated and its expression is displayed every few minutes, 1 h and 24 h.
DNA fragmentation test	DNA fragmentation is characteristic of apoptosis. After the cells are exposed to the peptide, the DNA is extracted and placed in agarose gel in order to visualize the DNA fragmentation.
TUNEL assay	It is an assay used for the detection of DNA fragments due to the process of apoptosis. This assay consists of the ability of terminal deoxynucleotidyl transferase to mark blunt ends of double-stranded DNA breaks independently of a template.
Anti-angiogenesis assay	Anticancer peptides are recognized for stopping angiogenesis caused by tumor cells. In this assay, venous endothelial cells from the human umbilical cord are used and confronted with the anticancer peptide. Then it is observed if there is a formation of blood connections or not compared with the control, expecting an inhibition of these by the anticarcinogenic peptide.
Flow cytometry	This test can determine whether or not there is cell or mitochondrial membrane damage, DNA fragmentation and cell cycle alteration. It also allows differentiation between necrotic, apoptotic or healthy cells.
Release of lactate dehydrogenase (LDH)	LDH is a cytoplasmic enzyme present in all cells and released into the cell space when the membrane ruptures. The assay uses the supernatant of the cells that were treated with the peptide by measuring the absorbance at 450 nm in microplates and relating the peptide concentration to the percentage of LDH release
Reactive oxygen intermediates (ROS) assay	This assay is used to detect the generation of ROS, whose generation induces damages in DNA, proteins, and membrane lipids. Kits such as the ROS assay kit (BestBio, Shanghai Co., China) are used which have a fluorescent probe that allows the intensity of the fluorescence to be detected by flow cytometry and directly correlated to an increase in ROS concentration

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
