# Peer review of "Peptides with Dual Antimicrobial–Anticancer Activity: Strategies to Overcome Peptide Limitations and Rational Design of Anticancer Peptides"

_molecules, 2020, doi:10.3390/molecules25184245_

Round 1

Reviewer 1 Report

This is a poorly written, and poorly organized review article. First off the bat, the title is misleading. There is no significant mention of peptides with dual antimicrobial and anti-cancer properties. The authors do not even get to the topic until almost half way through the review article. It starts with evolutionary concerns and meanders through history without getting to the punch line. It briefly mentions a dozen different techniques before getting to the in silico techniques. Either the title should be revised, or more detailed attention given to antimicrobial peptides, anti-cancer peptides, and than specific mention of the studies looking at dual-function peptides.

Reviewer 2 Report

This review aims to focus on peptides with dual antimicrobial and anticancer activities. Specifically, it aims to focus "on the current status of research" and the use of in silico methods. The manuscript covers many (historical) aspects of bacterial infections and cancer, as well as introducing the reader to general peptide pharmaceutical research. However, the authors only dive into the stated topic on page 13, or section 10 out of 13. I think it would be much more interesting to expand on these sections and to shorten the rest of the manuscript, as sections 1-9 have been reviewed elsewhere. In addition to this, there are a number of other points the authors need to address before this manuscript can be accepted for publication:

1) Many of the figures appear to be reproduced directly from the web, without any permission being obtained (e.g. Figure 1C). At the very least, statements need to be made that these graphs can be used freely.

2) Line 62: Most people agree that antibacterial development occured between 1930 and 1990 - the lipopeptides were discovered in 1987.

3) Please capitalize the word "gram" when referring to Gram-positive or negative bacteria. It is named after the person who invented this staining procedure to distinguish between the two types of bacteria, and hence should always be capitalized.

4) The data presented in Table 1 is from the CDC and not the WHO. In addition, reference 20 is not the source of this data.

5) The caption to Figure 2 is difficult to read and understand. Where is Table 19? The sentences are fragmented...

6) Line 112: The statement that "Consequently, the era of antibiotics could come to an end" is a bit imprecise. One can argue (and the authors have) that there has been little development since 1990, so one could say that the era is already at an end. Please use more precise wording.

7) Line 117: Lysine is an amino acid, not an enzyme so I am not sure what the authors are referring to here.

8) Figure 4: Why are some circles red and some blue? What is the purpose of the big circle in the middle?

9) Reference 37 is completely incorrect.

10) Line 234: I have never encountered secondary structures such as "lamina β" and "structures αβ". The authors should use the standard terms: α-helices, β-sheets, β-turns, random coil.

11) Table 4: What is the difference between the 1st and the 5th entry? Are they not the same?

12) Although I could understand why the authors may want to point out the situation in their home country, I am not sure that the paragraphs surrounding Figure 6 are appropriate for a Molecules review.

13) Section 8: It might be nice for the authors to include additional references here, as the book (ref. 51 and 59) may be difficult to come by. Examples of recent reviews include: Kumar et al., Biomolecules, 2018; Drayton et al., Molecules, 2020.

14) Line 338: The authors have to be careful in giving daptomycin as an example of lipidation. It comes in this form naturally, whereas the lipidation the authors are referrring to is done synthetically.

15) Figure 7 is difficult to understand. How do the solutions listed in green fix the problems in red?

16) Reference 42 could be replaced by a more recent one: e.g. Haney, Straus, and Hancock, Frontiers in Chemistry, 2019.

17) Doesn't the statement made on line 400 contradict the high production costs listed in Table 2. If costs are low specifically for anti-cancer peptides then this should be explained.

18) I am not sure what the purpose of section 11 is, in particular since there are other reviews which describe the methods in much more detail (e.g. Raheem and Straus, Frontiers in Microbiology, 2019). This section may be easier to read through as a table.

19) Section 12 is way too general. It would be better to describe in more detail how the various methods can be directly applied to antimicrobial/anticancer peptides, instead of describing the history of omics.

20) The authors have neglected to discuss a whole body of work which involves in silico designs. I am refererring here to a number of papers by E.F. Haney et al.

Overall the review lacks focus. It would be much easier to read if the authors described details about antimicrobial/anticancer peptides and in silico methods. There is enough material available for a good review. I would also suggest avoiding the generalizations (e.g. historical background, etc).

Other minor points:

  • Please correct the definition for MTT.
  • Line 519: beta leaves?

Round 2

Reviewer 1 Report

This is a much improved, more readable version of the manuscript. I would recommend acceptance and publish after minor spelling/grammatical mistakes are all ruled out.

Reviewer 2 Report

The authors appear to have addressed all the concerns raised by the reviewer. I recommend this paper be accepted for publication, once the editing is complete.